# The Effect of Particle Size and Surface Roughness of Spray-Dried Bosentan Microparticles on Aerodynamic Performance for Dry Powder Inhalation

**DOI:** 10.3390/pharmaceutics12080765

**Published:** 2020-08-13

**Authors:** Yong-Bin Kwon, Ji-Hyun Kang, Chang-Soo Han, Dong-Wook Kim, Chun-Woong Park

**Affiliations:** 1College of Pharmacy, Chungbuk National University, Cheongju 28644, Korea; yongbin6474@naver.com (Y.-B.K.); jhkanga@naver.com (J.-H.K.); hsoo0805@naver.com (C.-S.H.); 2Division of BT Convergence, Cheongju University, Cheongju 28530, Korea; pharmengin@gmail.com

**Keywords:** bosentan, inhalable dry powder, surface morphology, roughness, physicochemical properties, aerodynamic properties

## Abstract

The purpose of this study was to prepare spray dried bosentan microparticles for dry powder inhaler and to characterize its physicochemical and aerodynamic properties. The microparticles were prepared from ethanol/water solutions containing bosentan using spray dryer. Three types of formulations (SD60, SD80, and SD100) depending on the various ethanol concentrations (60%, 80%, and 100%, respectively) were used. Bosentan microparticle formulations were characterized by scanning electron microscopy, powder X-ray diffraction, laser diffraction particle sizing, differential scanning calorimetry, Fourier-transform infrared spectroscopy, dissolution test, and in vitro aerodynamic performance using Andersen cascade impactor^™^ (ACI) system. In addition, particle image velocimetry (PIV) system was used for directly confirming the actual movement of the aerosolized particles. Bosentan microparticles resulted in formulations with various shapes, surface morphology, and particle size distributions. SD100 was a smooth surface with spherical morphology, SD80 was a rough surfaced with spherical morphology and SD60 was a rough surfaced with corrugated morphology. SD100, SD80, and SD60 showed significantly high drug release up to 1 h compared with raw bosentan. The aerodynamic size of SD80 and SD60 was 1.27 µm and SD100 was 6.95 µm. The microparticles with smaller particle size and a rough surface aerosolized better (%FPF: 63.07 ± 2.39 and 68.27 ± 8.99 for SD60 and SD80, respectively) than larger particle size and smooth surface microparticle (%FPF: 22.64 ± 11.50 for SD100).

## 1. Introduction

Bosentan hydrate (BST) is an endothelin-A and -B receptor antagonist. BST reduces lung and systemic vascular resistance and thus is used in pulmonary artery hypertension (PAH) [1,2]. BST was approved under the trade name Tracleer^®^ by the US Food and Drug Administration, and it is orally administered to treat people with moderate PAH [3]. However, the oral dosage of BST has systemic circulation side effects. The adverse effects of BST include liver damage via the hepatic metabolism, embryo-fetal toxicity, pulmonary veno-occlusive disease, and decreases in right coronary blood flow [4,5]. Therefore, to reduce systemic side effects, a drug delivery system that can act on only in the pulmonary circulation is needed [6]. As an alternative approach, targeted lung delivery through dry powder inhalers (DPIs) avoids the first pass hepatic metabolism and reduces systemic circulation, thus providing higher drug concentrations in the target area and limiting side effects.

The delivery of dry powders to the lung has become a popular delivery system for both topical and systemic disease states, due to their large surface area, thin epithelial barrier, and rich blood supply [7]. Pulmonary delivery systems are classified into three types: DPIs, pressurized metered-dose inhalers (pMDIs), and nebulizer. DPIs enable delivering high doses to the airways compared to other inhalation devices [8]. The aerosol performance of the particles is influenced by physicochemical properties such as particle size distribution, particle shape, surface roughness, interparticle force, and solid state, which determines the efficiency of DPI. In addition, microparticles with 1–5 µm in aerodynamic size are needed for the pulmonary delivery [9,10].

Microparticles can be prepared by spray drying technique and have been successfully used for the production of microparticle delivery systems [11,12,13,14,15,16]. This technique provides the advantage of converting the drug solution into dry particles with a narrow particle size distribution, and the dry particles can be tailored by changing the spray-drying parameters [17,18].

In this study, we prepared formulations for DPIs containing BST to deliver the drug to the bronchioles and alveolae. To deliver BST to the deep lung, the particle size must be micronized to respirable. However, the micronized particles with small particle size and high surface energy are highly agglomerated and poorly dispersed [19].

To overcome the cohesiveness of micronized particles for inhalation, mixing with relatively large coarse lactose or changing the surface morphology is needed. The effect of surface morphology on increasing aerosol performance was explored using nonporous corrugated solid particles [20]. These particles exhibit better aerodynamic performance than smooth spherical particles. In addition, the lower cohesion force between the corrugated particles enables them to de-agglomerate more easily on impaction [21]. The dispersion behavior of the corrugated particles is less dependent on the inhaler device and airflow. The surface roughness, particle size, and morphology of the particle directly influenced aerosol dispersion [22]. Particles with corrugated (also called wrinkled) surface lower the inter-particulate interactions between particles and increase the fine particle fraction (FPF) from DPIs [23,24,25]. However, in most studies, the aerosol performance of inhalable microparticle was evaluated only by aerodynamic particle size distribution (ASPD) using a cascade impactor test such as ACI, NGI, or MSLI without directly confirming the actual movement of the aerosolized particles.

In this study, a particle image velocimetry (PIV) system was introduced to confirm the actual movement of BST particles. PIV is a useful experimental tool for the characterization of flow properties in fluids. The fluid with entrained particles is illuminated by the incident laser. The motion image of the seeding particles is used to calculate parameters such as X-axis and Y-axis velocity and direction of the flow. Few studies have focused on the effects of the movement of particles in aerosol and the PIV system can directly observe particle aerosol flow. The aim of this study was to prepare BST microparticles with varying particle size, shape, and surface morphology from smooth to moderately corrugated, and to determine whether the particle size distribution, shape, and raisin-like (or high roughness) surface positively affect the aerosol performance of spray-dried particles by ACI and a PIV system.

## 2. Materials and Methods

### 2.1. Materials

Bosentan hydrate (C_27_H_31_N_5_O_7_S; molecular weight (MW): 569.63 g/mol) was purchased from Xi’an Sgonek Biological Technology Co. LTD. (Xi’an, China). Ethanol, acetonitrile, and all other reagents were HPLC or analytical grade.

### 2.2. Preparation of Spray-Dried Bosentan Microparticles

The BST microparticles were prepared using spray-drying method with a laboratory scale spray dryer (EYELA SD-1000, Rikakikai Co. Ltd., Tokyo, Japan). As shown in Table 1, 3.5 g of BST were dissolved in different concentration (60%, 80%, and 100%) of ethanol solutions (0.7% *w*/*w*) and spray-dried with the following parameters: inlet temperature of 110 °C, nozzle size of 0.4 mm (two-way nozzle spray type), feeding rate of 6 mL/min, atomization air pressure of 240 kPa, and drying air flow rate of 0.24 m^3^/min. The amounts of SD formulations powder obtained after the spray drying process were 161, 54, and 158 mg for SD60, SD80, and SD100, respectively.

### 2.3. Particle Size and Size Distribution

The particle size and particle size distribution of the raw BST and spray-dried BST microparticles was determined using a laser diffraction particle sizing by Mastersizer 2000 (Malvern Instruments, Worcestershire, UK). After dispersing the samples in non-swelling solvent, isopropyl alcohol (5% *w*/*v*), BST microparticle size distribution analysis was carried out by a wet dispersion method.

### 2.4. Scanning Electron Microscope (SEM)

The morphology of the spray-dried BST microparticles was confirmed by SEM (ZEISS-GEMINI LEO 1530, Zeiss, Oberkochen, Germany) [26]. The raw BST and spray-dried BST microparticles were spread on a carbon tape, and then unattached microparticles were blown off and platinum coated to a thickness of 200 Å using a Hummer VI sputtering device. The magnifications of 10,000× and 50,000× and a voltage of 3 kV were used.

### 2.5. Differential Scanning Calorimeter (DSC)

DSC instrument (DSC 2910, TA instruments, New castle, DE, USA) was used to analysis the enthalpy changes of the raw BST and spray-dried BST microparticles. All samples were placed in an aluminum pinhole pan to prevent residual humidity present in the powder and then heated from 30 to 180 °C at a scanning rate of 10 °C/min. Nitrogen was used as a DSC standard purge gas due to its better heat conductivity.

### 2.6. Powder X-ray Diffraction (PXRD)

The PXRD of the all samples was performed using an X-ray diffractometer (XDS 2000, SCINTAG, Waltham, MA, USA) with a Cu radiation source (40 kV, 40 mA). The scanning range of 2*θ* was taken from 5° to 50°, using a step size of 0.009°/2*θ* at ambient temperature.

### 2.7. Fourier Transform Infrared (FT-IR) Spectroscopy

FT-IR analysis was performed using the FT-IR spectroscopy (IFS 66 v/S, Bruker Optics, Ettlingen, Germany) using the potassium bromide technique and a deuterated triglycine sulfate (DTGS) detector. Spectra were collected in the 4000 to 400 cm^−1^ range at a resolution of 4 cm^−1^ and 16 scans.

### 2.8. In Vitro Dissolution Study

To determine the sink conditions, solubility test of raw BST in the dissolution medium was conducted. An excess amount of BST (100 mg) was added to 10 mL of phosphate buffered saline (PBS) buffer with Tween 80 (5%, *w*/*w*) in glass vials and vortexed for 20 s. Mixtures were equilibrated for 48 h at room temperature and centrifuged for 10 min at 2500 rpm. The supernatants were removed and filtered through a 0.45-µm membrane. The in vitro dissolution profiles of spray-dried formulations and raw BST were evaluated via a Franz diffusion cell system (FCDS-900C, Labfine Instruments, Anyang, Korea). Franz diffusion cell (Lab-fine, Seoul, Korea) was used to mimic the diffusion-controlled air–liquid interface of the lung after inhalation of the drug [27]. The Franz cell reservoir was filled with PBS (pH 7.4) containing Tween 80 (5%, *w*/*w*) and maintained at 37 °C. The capacity of the receptor chamber was 12 mL and the inner diameter was 12 mm. The medium was continuously stirred to ensure homogeneity. Mixed cellulose ester membrane filter with a pore size of 0.45 µm and a diameter of 25 mm (Advantec, Tokyo, Japan) was used as a barrier, placed in the holder to allow contact with the medium and had no liquid in donor compartment. Five milligrams of precisely weighted raw BST and SD formulations were placed on the membrane at the air–liquid interface. After dissolution of predetermined time point (10, 20, 30, 60, 120, 240, 480, and 720 min), 200 µL of samples were removed and 200 µL of fresh buffer were filled to maintain a constant medium volume. The removed samples were centrifuged for 10 min at 1000 rpm and the supernatants were removed and analyzed by HPLC method. The dissolution data were fitted to a first-order kinetic model to investigate the drug dissolution mechanism as the following equation.

First-order kinetic model:(1)$$\ln\left[A\right]=\ln{\left[A\right]}_{0}-k_{1}t$$

[A] is the amount of drug dissolved at time t, [A]_0_ is initial amount of drug in medium, and $k_{1}$ is the first-order constant.

In addition, comparison between the dissolution rates was confirmed using mean dissolution time (MDT) as following equations:(2)$$MDT=\frac{\sum_{i=1}^{n}t_{mid}∆M_{i}}{∆\sum_{i=1}^{n}∆M_{i}}$$
where ∆M_i_ is the fraction of drug released in time t_i_ (calculated by (t_i_ + t_i − 1_)/2) and i is the sample number.

All experiments were done with n = 4.

### 2.9. Particle Image Velocimetry (PIV)

PIV was used to visualize the actual movement of the spray-dried BST in air flow. Ten milligrams of spray-dried BST were placed in a Size 4 hydroxypropyl methylcellulose (HPMC) hard capsule for loading into a Handihaler^®^ device (Boehringer Ingelheim, Biberach, Germany). As shown in Figure 1A, to visualize the actual movement of the microparticles, a Handihaler^®^-mounted mouthpiece was inserted into a transparent acrylic chamber size of 120 mm × 90 mm × 90 mm. After confirming the flow rate of 60 L/min using a flow meter (DFM 2000, COPLEY Scientific, Nottingham, UK), the aerosolization was actuated using a pump (TISCH environmental, Inc., Cleves, OH, USA). The grid design is shown in Figure 1B, setting the X-axis grid to 45 (real distance is 120 mm) and the Y-axis grid to 30 (real distance is 90 mm). The angular distribution and X–Y velocity plot of the particle flow field were analyzed for 100 ms in the center of the chamber at grid point (15, 15) (X-axis grid and Y-axis grid, respectively), as shown in Figure 1B.

The diode Pumped Green 532 nm laser was positioned so that the laser sheet was parallel to the mouthpiece and perpendicular to a high-speed camera (HAS-D71M, Ditect Corporation, Tokyo, Japan). Videos of aerosolized microparticles were recorded at 8000 frames per second (1 frame/0.125 ms) and the image was set to 640 × 480 pixels. All data related to PIV were used in image analysis software (Flownizer 2D, Ditect corporation, Tokyo, Japan). PIV measurements were performed five times, and all PIV images are representative. The color scale means X-axis and Y-axis vector velocimetry. The blue and red colors mean that vector velocimetry is 0 and 10 mm/ms, respectively.

### 2.10. In Vitro Aerosol Performance Using Andersen Cascade Impactor

Andersen Cascade Impactor (ACI) was used to evaluate the in vitro aerosol performance of spray-dried BST microparticles according to USP Chapter <601>. The flow rate of 60 L/min and the 4 kPa pressure drop of the inhaler were checked using a flow meter and a critical flow controller (COPLEY scientific, Nottingham, UK) and then actuated using a pump at a flow rate of 60 L/min for 4 s. A Handihaler^®^ with a capsule loaded with 10 mg of sample was inserted into the mouthpiece connected to the induction port. To analyze the BST remaining in the capsule and deposited on plate, they were dissolved in 15 mL of the mobile phase and measured using HPLC. ACI Stages 1–6 have an aerodynamic cut-off diameter of 8.6, 6.5, 4.4, 3.3, 2.0, 1.1, 0.54, and 0.25 µm, respectively. The emitted dose (ED) was used as the amount of dry powder released from the HPMC hard capsule. The fine particle fraction (FPF) is the fraction of particles with an aerodynamic size of 4.4 µm or less in administered dose. The ED and FPF were calculated using the following equations:(3)$$\mathrm{Emitted}\;\mathrm{dose}\;\left(\mathrm{ED}\right)\%=\frac{Initial\;mass\;in\;capsule-Final\;mass\;remaining\;in\;capsule}{Initial\;mass\;in\;capsule}\times 100$$
(4)$$\mathrm{Fine}\;\mathrm{particle}\;\mathrm{fraction}\;\left(\mathrm{FPF}\right)\%=\frac{Mass\;of\;particles\;on\;stages\;1\;through\;filter}{Total\;mass\;on\;all\;stages}\times 100$$

The mass median aerodynamic diameter (MMAD) is defined as the diameter of the particle with 50% larger and 50% smaller reaching the impactor, excluding particles deposited on the throat. Geometric standard deviation (GSD) measures the dispersion of particle diameter [28].

### 2.11. HPLC Analysis Method

The bosentan monohydrate was analyzed by HPLC (Ultimate 3000 series HPLC system; Thermo Fisher Scientific, Waltham, MA, USA) using a Luna L11 250 mm × 4.60 mm, 5 µm column (Phenomenex, Torrance, CA, USA). The mobile phase consists of 60:40 (*v*/*v*) acetonitrile and buffer (pH 2.5 with 0.1% *v*/*v* trimethylamine). The mobile phase flowed at a rate 1.5 mL/min. The column temperature was set to at 35 °C, and the injection volume was 10 µL. Six points (1.25, 2.5, 5, 25, 50 and 100 µg/mL) were considered and the required BST concentration range for the analysis was 1–25 µg/mL.

### 2.12. Statistics

One-way analysis of variance (ANOVA) and Tukey’s post hoc test (SPSS, version 22.0, Chicago, IL, USA) were used to evaluate the differences. A *p* value less than 0.05 was considered to be statistically significant.

## 3. Results and Discussion

### 3.1. Particle Morphology

Scanning electron microscopy images of raw BST and spray-dried BST microparticles are shown in Figure 2. The raw BST had irregular, non-spherical morphology (Figure 2G). SD60 had irregularly folded and flocculated round morphology with rough surface and monodisperse size range (Figure 2E,F). SD80 had spherical morphology with a rough surface and monodisperse size range (Figure 2C,D). SD100 had spherical morphology with a smooth surface (Figure 2A,B). The difference in particle morphology among spray-dried formulations was a result of different ethanol concentration in feeding solution. This indicated that evaporation rate during the drying process determined the particle morphology [29].

### 3.2. Physicochemical Properties

#### 3.2.1. Particle Size Distribution

Depending on the target delivery area of the DPIs, a particle has to be small enough, preferably 1–5 µm in aerodynamic diameters, for deep lung delivery [9,10,30]. The aerodynamic diameter of a particle, which is defined as that of sphere of unit density (1 g/cm^3^), generally depends on particle characteristics (e.g., particle shape, morphology, density, and geometric size) and air flow, thus a small aerodynamic diameter can be obtained from volumetric equivalent diameter, low density particles, and non-spherical particles [20,31]. The particle size and size distribution of spray-dried BST microparticles and raw BST are shown in Table 2. Spray drying process reduced the particle size of the BST with narrow size distribution.

#### 3.2.2. DSC

As shown in Figure 3, DSC thermograms of spray-dried BST microparticles and raw BST were investigated to characterize their thermal behavior. The raw BST showed endotherm peak at 111.6 °C. The peak at 111.6 °C represents the melting point of BST, indicating that a high crystalline solid state is achieved [32] and the peak of the dehydration of the hydrated form was overlapped by melting peak. However, the solid state of SD formulations was extremely transformed by spray drying. The endotherm peak of the drug was not observed in SD100, SD80, and SD60, which indicates that the crystalline solid state of the drug was transformed to amorphous (XRD data are shown in Appendix A). During spray drying, the fast evaporation of the solvent may result in insufficient crystallization time [33,34]. There was no difference in DSC thermograms among spray-dried BST microparticles.

#### 3.2.3. FT-IR Spectroscopy

As shown in Figure 4, FT-IR spectra of spray-dried BST microparticles and raw BST were investigated to characterize their functional groups. Compared to the raw BST, the FT-IR spectrum of the SD formulations exhibited a disappearance in the peaks at ~3650–3600 cm^−1^, which corresponds to the O–H stretching. The O–H stretching was derived from the monohydrate of BST monohydrate and was not observed in the SD formulation by dehydration during the spray drying process. FT-IR spectra distinguishing spray-dried BST microparticles are present at ~3000–2850 cm^−1^ corresponding to C–H stretching. It is also not changed after spray drying process and there are no significant shifts in the positions of peaks. There are few variations in the intensity in at 1080 cm^−1^ (C–O stretching) and 1300 cm^−1^ (S=O stretching) [35], which is due to the crystallographic difference between amorphous SD formulations and typical crystalline BST monohydrate [36].

As expected from the DSC and PXRD, FT-IR spectra of the spray-dried BST microparticles showed the absence of chemical changes after spray drying process.

### 3.3. In Vitro Drug Dissolution Study

The solubility result of raw BST in dissolution medium was 1345.3 ± 0.032 µg/mL. Therefore, 12 mL of dissolution medium maintain the sink conditions for the dissolution test of 5 mg raw BST. The dissolution profiles of spray-dried BST microparticles and raw BST were investigated by Franz diffusion cell (Figure 5). The SD100, SD80, and SD60 showed an initial burst up to 1 h compared with raw BST after starting the dissolution test. In particular, SD100, SD80, and SD60 showed higher values than raw BST during 8 h. Amorphous compounds have higher solubility and faster dissolution rate than crystalline [37]. After the dissolution of 12 h, no significant difference was observed in dissolution profiles in each microparticle. The first-order processes depend only on the dissolved substance concentration. Thus, substances with higher solubility dissolved more quickly and can diffuse through the membrane compared to substances with lower solubility [38]. The kinetic of drug dissolution was calculated according to the first order. Table 3 shows the correlation coefficients (r^2^) and dissolution constant (k_1_). To compare the initial dissolution rate, the first-order kinetic was applied for 1 h after dissolution. The k_1_ value of raw BST was significantly lower than other SD formulations (*p* < 0.01, ANOVA/Tukey). In addition, the comparison between the dissolution rates was confirmed using the mean dissolution time (MDT). The MDT value was 5.82, 3.69, 3.31 and 3.91 h for raw BST, SD100, SD80, and SD60, respectively. It showed significant difference in raw BST (*p* < 0.05, ANOVA/Tukey). It means that the amorphous solid state improves the dissolution rate in early stage. The rate of dissolution was lower than 30%, but the difference between raw BST and SD formulation for 1 h was clear.

### 3.4. Particle Image Velocimetry (PIV)

DPI is a dosage form for delivering drug powder to respiratory tract and lung. Particle size and size distribution, particle morphology, surface roughness, fine particle fraction, flow rate, cohesion, and adhesion force have been reported to affect the process of aerosolization. Particle dispersion is also associated with impaction, shear stress, and turbulent flow [39]. To identify the particle dispersion dynamics, several studies have been carried out using approaches such as computational fluid dynamics (CFD) [40,41] and PIV system [42,43]. In this study, PIV was used to provide experimental observation of the particle dispersion process. As a result of confirming high-speed video imaging, there was a difference depending on the shape, surface roughness, and size distribution of the particles.

Figure 6 shows the particle dispersion properties of the BST formulations as examined by PIV system, and the vector analysis at 25, 50, 75, and 100 ms after emission were obtained at the side cross-plane. The bottom of the image is where the dry powder inhaler is located, the color of the arrow indicates the velocity of the particle, and the direction of the arrow indicates the direction of particle motion. Regarding the morphology of the BST formulations, the PIV results show very different particle velocity and flow fields. In SD100, the flow of the particles was dispersed rapidly only in the forward direction. In contrast, SD80, and SD60 showed lower velocities and more complicated directions than SD100.

Figure 7 shows the contour map of the particles at 25, 50, 75, and 100 ms after emission. SD100, SD80, and SD60 reached the end of the chamber after 360, 390, and 500 frames (Appendix A). Since 8000 frames were recorded per second, the time to reach the end of the chamber was 45, 48.75, and 62.5 ms for SD100, SD80, and SD60, respectively. In addition, the length of the X-axis was 120 mm, and the X-axis velocity was 2.67, 2.46, and 1.92 mm/ms for SD100, SD80 and SD60, respectively. At the 100 ms contour image, SD100 shows high velocity but SD60 and SD80 did not, confirming that the velocities of particles with large size were higher than particles with small size [44].

Figure 8 shows the X–Y velocity and angular distribution for 100 ms at the front point of the chamber for each of the formulations. In the X–Y velocity plot, SD100 has a relatively high X-axis velocity and shows Y-axis velocity between −1 and 1 mm/ms, while SD80 and SD60 have a low X-axis velocity and show the wide range of Y-axis velocity between −1 and 3 mm/ms. Particle angles of SD80 and SD60 were more broadly distributed than SD100. Therefore, SD80 and SD60 have more complex flow with various directions than SD100.

Hassan [45] reported that the rough surface would cause a slightly higher shape factor than spherical particles. In addition, Davies [46] reported that pollen shaped particles had a slightly higher shape factor than smooth surface particles. Therefore, the shape factor of the SD100 formulation with a spherical and smooth surface is expected to be the lowest. For the movement direction, SD100, which has a smooth surface, exhibits monotonous flow in only the frontal direction. In contrast, SD80 and SD60 have rough surfaces and a high levels of dynamic shape factor, hence particles generally disrupt air flow to the greatest extent. The complex flow and a large pressure differential across the particle cause particles to be suspended and generated turbulent flow at relatively low and variable velocities [45]. In addition, the surface asperities of the particles could lower the true area of contact between the particles, and thus reduce the powder cohesiveness [20]. These characteristics for SD80 and SD60 represent complex particle motion in comparison with SD100, which could improve the aerosol performances.

### 3.5. Andersen Cascade Impactor

Aerosol dispersion performance of the spray-dried BST microparticles was carried out using an Andersen cascade impactor and the Handihaler^®^ as dry powder inhalation device. MMAD, GSD, ED and FPF values obtained by ACI deposition study are reported in Table 4. The Figure 9 presents the distribution of BST as the percent deposition and deposited amount of each stage. SD60 and SD80, having smaller size than SD100, showed synergy in the performance of aerosol dispersion performance. In addition, the dispersion characteristics of particles in air flow are also defined by the balance of aerodynamic stress and particle aggregation strength [28]. Rough surface types of particles have smaller aggregation strength than the smooth particles due to the decreased contact area and inter-particulate forces [17]. Therefore, SD80 and SD60 showed higher FPF values and smaller MMAD than SD100 and raw BST, indicating that the SD80 and SD60 particle size and morphology are more suitable for drug delivery to deep lung. The %FPF values were 6.74 ± 0.95, 21.04 ± 11.75, 59.47 ± 10.46, and 59.97 ± 5.45 for raw BST, SD100, SD80 and SD60, respectively. The FPF% of the raw BST and the SD100 formulation was significantly lower than those of SD80 and SD60 (*p* < 0.005, ANOVA/Tukey). The MMAD values for SD100, SD80, and SD60 were 6.95 ± 0.63, 1.27 ± 0.05, and 1.27 ± 0.09 µm, respectively. The GSD values for SD100, SD80, and SD60 were 1.31 ± 0.63, 1.70 ± 0.08, and 1.65 ± 0.11, respectively. However, the raw BST showed a deposited ratio close to 90% at Stage −1, thus the MMAD and GSD values could not be calculated. Particles with small size and high roughness disperse better than large size and smooth surface particles [20,47]. Therefore, the strategy to improve the aerosol performance of powders was to create particles with small size and high roughness.

Eventually, SD100 impacted and was deposited at the upper stages of ACI; therefore, the values of FPF%, MMAD and GSD were significantly limited. However, SD80 and SD60 with small size and rough surface generally disrupted the air flow to a greater extent, resulting in a complex and slow flow, causing the prevention of inertial impaction.

## 4. Conclusions

In this study, BST microparticles (SD60, SD80, and SD100) were successfully prepared by spray drying method and physicochemical characteristics were investigated. Each BST microparticles showed different particle size, shape, and surface morphology, which had different effects in particle dispersion dynamics and aerosol performance. BST microparticles prepared by spray drying had amorphous solid state, which increased the solubility of the drug and contributed to the initial burst in the early stage of dissolution. The actual particle behavior in air stream was affected by particle size and surface roughness and was confirmed by PIV. In addition, aerosol performance was confirmed by ACI that SD80 and SD60 with small particle size and high roughness would produce an improved deposition in the deep lung region.

In most previous studies [17,20,48], the improvement of dispersion of the particles due to the roughness of the surface was explained by the reduction of the contact area and inter-particulate force, but this study tried to observe the actual movement of the particles directly through PIV. In conclusion, it was confirmed that the difference in the actual movement of particles in the air flow affects aerosol performance. Further studies are needed to explore the spray dried-BST microparticles’ therapeutic effects for PAH in animals.

## Figures and Tables

**Figure 1 pharmaceutics-12-00765-f001:**
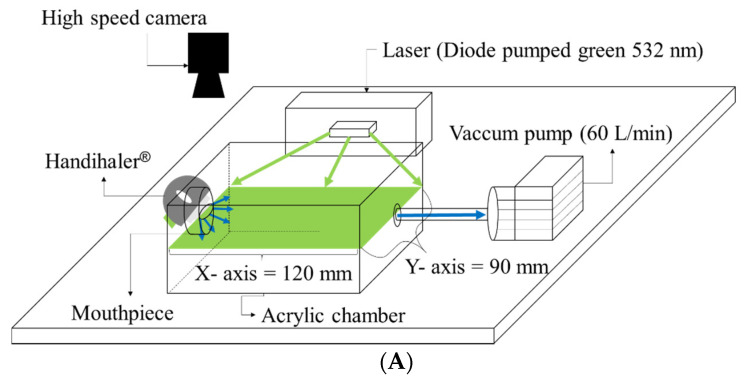
(**A**) Schematic diagram of PIV system; and (**B**) grid design and vector analysis point (15, 15).

**Figure 2 pharmaceutics-12-00765-f002:**
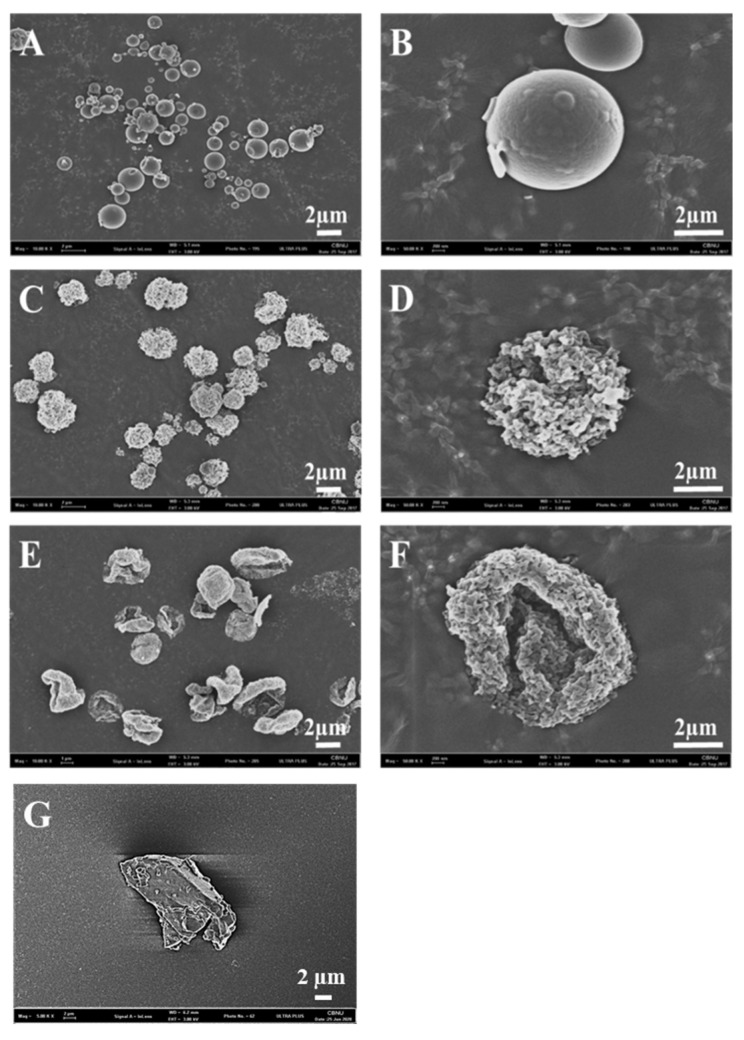
Scanning electron microscopy images of: (**A**) SD100; (**C**) SD80; (**E**) SD60; and (**G**) raw bosentan at magnification of 10 K; and (**B**) SD100; (**D**) SD80; and (**F**) SD60 at magnification of 50 K.

**Figure 3 pharmaceutics-12-00765-f003:**
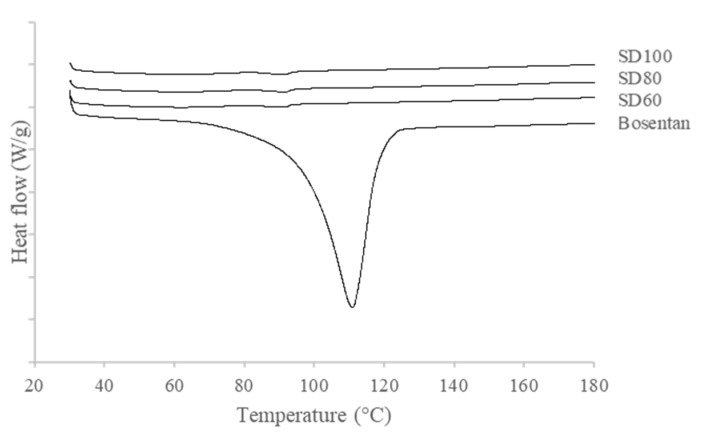
DSC thermograms of spray-dried bosentan microparticles and raw bosentan.

**Figure 4 pharmaceutics-12-00765-f004:**
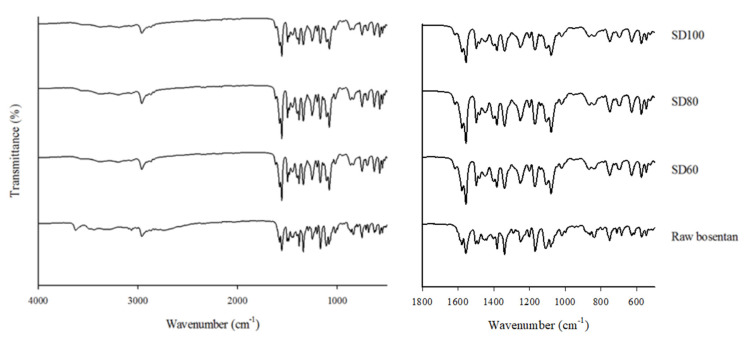
FT-IR spectra of spray-dried bosentan microparticles and raw bosentan.

**Figure 5 pharmaceutics-12-00765-f005:**
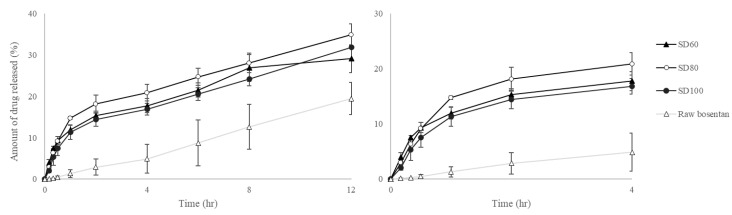
Dissolution profiles of spray-dried bosentan microparticles and raw bosentan in Franz diffusion cell.

**Figure 6 pharmaceutics-12-00765-f006:**
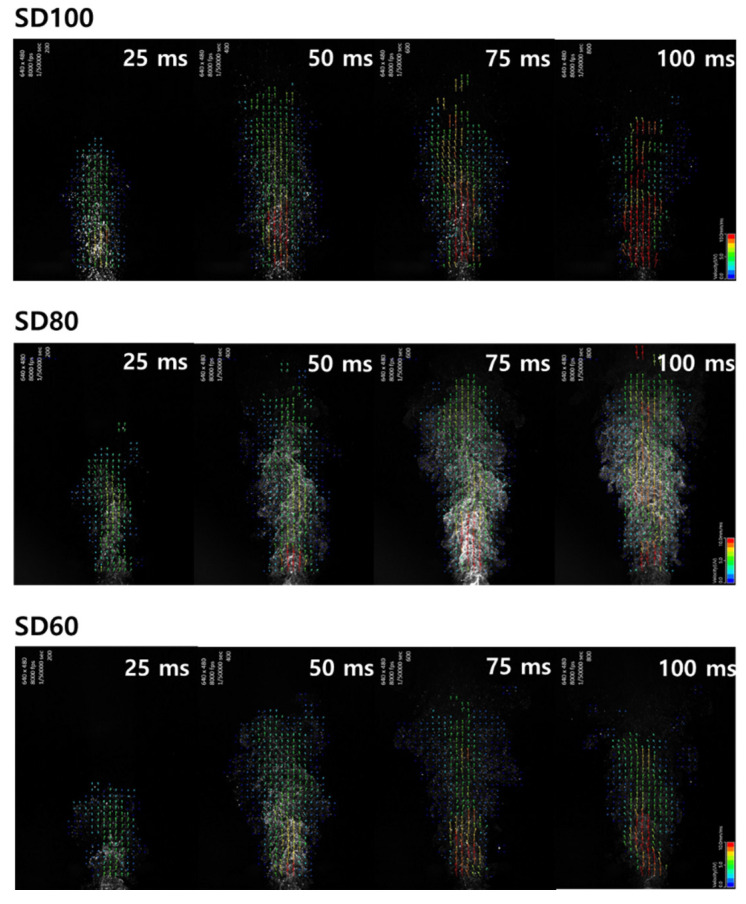
Vector images of particle flow field emitted from DPIs.

**Figure 7 pharmaceutics-12-00765-f007:**
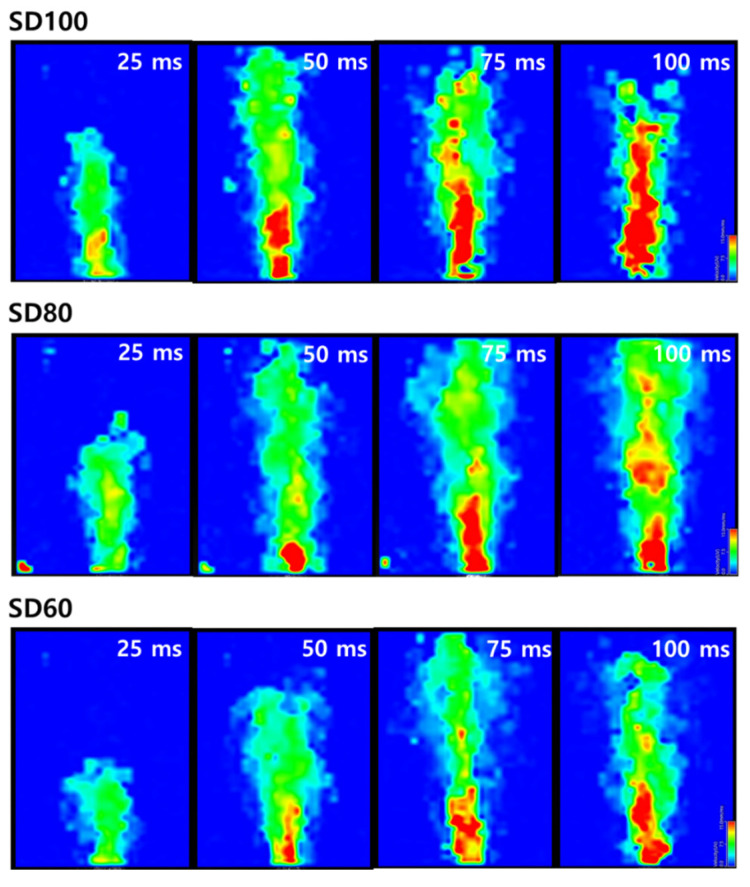
Contour images of particle flow field emitted from DPIs.

**Figure 8 pharmaceutics-12-00765-f008:**
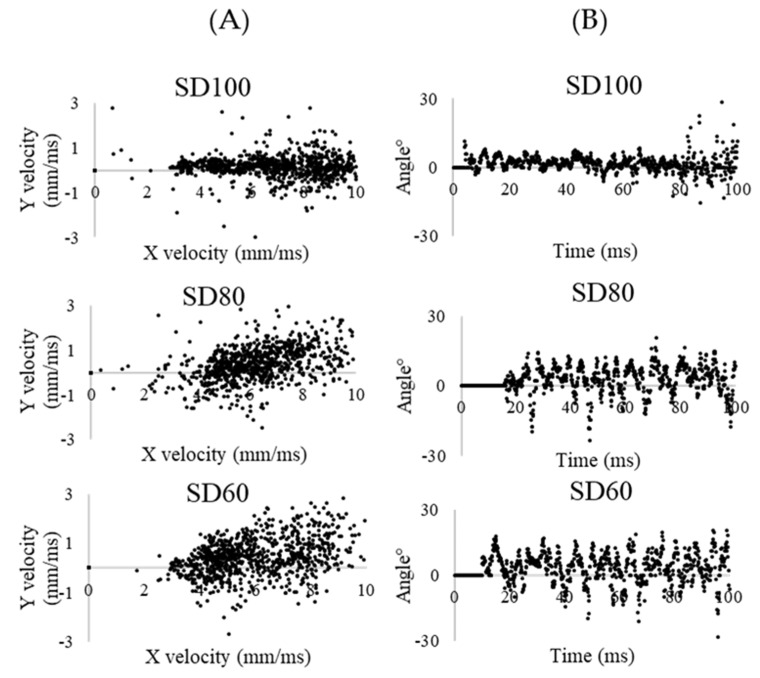
(**A**) X–Y velocity distribution of each formulations; and (**B**) angular distribution of each formulation.

**Figure 9 pharmaceutics-12-00765-f009:**
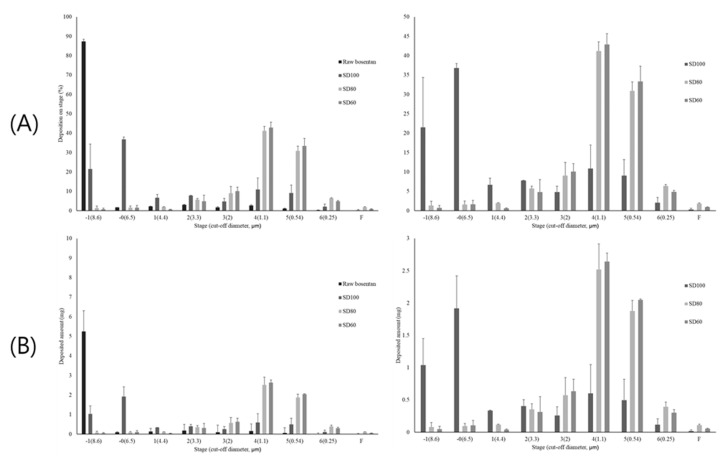
Aerosol performance in each stage of ACI for raw BST and spray-dried BST microparticles (mean ± standard deviation, n = 3): (**A**) deposited percentage; and (**B**) deposited amount (mg).

**Table 1 pharmaceutics-12-00765-t001:** Formulation and spray drying parameters of the spray-dried bosentan microparticles.

Formulation	Bosentan Concentration	Ethanol Concentration	Temperature	Feeding Rate	Atomization Pressure	Drying Air Flow Rate	Yield
	(*w*/*w*)	(*w*/*w*)	(°C)	(ml/min)	(kPa)	(m^3^/min)	(%)
SD60	0.70%	60%	110/55 *	6	240	0.24	4.60
SD80	0.70%	80%	110/58 *	6	240	0.24	1.54
SD100	0.70%	100%	110/60 *	6	240	0.24	4.51

* Inlet/outlet temperature.

**Table 2 pharmaceutics-12-00765-t002:** Particle size and size distribution of raw material and spray-dried formulations.

	Raw Bosentan	SD60	SD80	SD100
D_v_ (10, µm)	2.46	1.33	1.68	3.81
D_v_ (50, µm)	11.3	4.76	4.88	6.94
D_v_ (90, µm)	20.7	8.78	8.52	10.8
Span	1.619	1.567	1.401	1.008

**Table 3 pharmaceutics-12-00765-t003:** First-order kinetic and MDT of SD formulations and raw bosentan in a Franz diffusion cell.

Formulation	First Order		MDT (Hour)
	r^2^	k_1_	
SD100	0.959 ± 0.039	0.121 ± 0.019	3.69 ± 0.57
SD80	0.982 ± 0.012	0.160 ± 0.005	3.31 ± 0.91
SD60	0.873 ± 0.038	0.122 ± 0.012	3.91 ± 0.14
Raw bosentan	0.953 ± 0.041	0.014 ± 0.002	5.82 ± 0.62

**Table 4 pharmaceutics-12-00765-t004:** Aerosol performance parameters of raw BST and spray-dried BST microparticles (mean ± standard deviation, n = 3).

Formulation	Aerosol Performance Parameters			
	ED (%)	FPF (%)	MMAD (µm)	GSD
Raw bosentan	96.90 ± 0.87	6.74 ± 0.95	NaN	NaN
SD60	94.96 ± 5.15	59.97 ± 5.45	1.27 ± 0.09	1.65 ± 0.11
SD80	86.76 ± 4.26	59.47 ± 10.46	1.27 ± 0.05	1.70 ± 0.08
SD100	94.23 ± 5.06	21.04 ± 11.75	6.95 ± 0.63	1.31 ± 0.63

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
