# Peer review of "The Effect of Particle Size and Surface Roughness of Spray-Dried Bosentan Microparticles on Aerodynamic Performance for Dry Powder Inhalation"

_pharmaceutics, 2020, doi:10.3390/pharmaceutics12080765_

Round 1

Reviewer 1 Report

The work from Kwon Yong-Bin and co-workers has been revised according to the comments provided by the reviewers. Briefly, the concerns the preparation and the characterization of Bosentan hydrate (BST) microparticles by spray-drying, in order to obtain a formulation that could potentially be used for the treatment of moderate pulmonary arterial hypertension. Three formulations have been proposed and their physico-chemical properties was in vitro characterized. As the previous version of the manuscript, the work maintains a good design of each experiments and the results and discussion section was significantly improved. The main structure where each in vitro characterization is separately presented and commented still remains, but the results are better correlated, and the reader is almost naturally conducted to the conclusions of the work. This is really appreciated. However, Authors would like to consider again the following suggestions:

  • Line 72-73: this sentence should be rephrased since it seems to miss a conclusion such as “[…] mixing with relatively large coarse lactose or changing the surface morphology is needed”.
  • Line 91: “the aim of this study” is missing.
  • Table 1: it should be better to report the yield of the spray drying process as percentage not as a mass. The information of the spray-dried BST mass can be included in the text to give an idea of the dimension of the batch scale production.
  • Paragraph 2.3: which was the particle concentration of each sample analysed by laser diffraction? The particle-particle interaction should be carefully considered to obtain reproducible and reliable results in the size analysis.
  • Paragraph 2.5: did the Authors performed only one thermal scan in the DSC analysis? Under which atmospheres the DSC analysis was conducted?
  • Line 146: maybe it would be easier to report the smaller time points not in hour but in minutes.
  • Figure 2: a plot of the calibration curve was not necessary, as probably more than one calibration curve was used over all the experiments performed. The main information asked in the previous revision of the work is now reported in the text (i.e. BST concentration range).
  • Line 241: if Authors want to report a theoretical approach via Peclet number consideration, they should then make a comparison with the comments provided for the particle morphology. Again, the paragraph related to Peclet number still remains confused and not providing a significant value to the results discussion.
  • Line 261: maybe only the size distribution of formulation SD100, with a span value of 1.008, can be really considered as narrow. I feel to suggest to the Authors to deepen and differentiate the comment about the particle size distribution.
  • Line 261-266: this sentence here has no sense at this stage of the manuscript where size distribution results are discussed.
  • Table 2: are there any standard deviation for the size results reported in Table 2?
  • Line 300: this sentence still remains confusing. It is clear from the results that no chemical modification occurred in BST after the spray-drying, but which are those “characteristics peaks” here mentioned?
  • Line 306: “after starting the dissolution test” has been already mentioned in line 305, and it is obvious that BST is release after starting the dissolution, not before. I suggest to the Authors to delete it in line 306.
  • Line 307: I suggest to support this sentence performing a solubility study of raw BST and spray-dried formulations. Authors should also consider that the dissolution rate of a molecule is affected by the particle size of the powder: BST raw material has all the diameter reported in Table 2 higher than the microparticles and this would contribute to the different burst effects together with the different solubilities (crystalline vs solid state). So, I suggest to check this point with a solubility study.
  • Line 315-316: significant different compare to microparticle formulations?
  • Table 3: are there any standard deviation for the first order parameters?
  • Figure 7: y-axis maximum value can be reduced since only the 40% of BST has been dissolved. I also suggest to provide an enlargement of the earlier time points to better highlight the differences described in the text for the burst effects. Why the dissolution study was prolonged only for 12h? Did the Authors consider to prolong the study over 12h to assess if more than the 40% of BST can be dissolved?
  • Figure 8: which formulation is reported in the PIV images here? No information is provided.
  • The same of the previous point for Figure 9.
  • Line 379: the acronyms MMAD, GSD and ED have never described in the manuscript.
  • Figure 11: I feel to suggest to the Authors to disrupt the y-axis to highlight the lower results.
  • Line 391-392: the text in these lines should be put above Figure 11.
  • Line 419: the acronym SD-BST has never been used and, thus, explained.
  • The stability of the spray-dried formulations has not been mentioned and commented, despite the long-term stability is a point that must be studied and assessed. Would the Authors want to provide a comment on that?
  • Throughout the manuscript there are some typo errors (e.g. lines 136, 234, 243, 351, 384).

Reviewer 2 Report

This manuscript reports 3 spray dried formulations of bosentan particles. SEM, DSC, FTIR, dissolution, PIV, and ACI were used to characterize the particles produced. While the characterization tests performed are quite comprehensive, the specific aim of the study is not clear. Particularly, the manuscript title “effect of surface morphology” is not being demonstrated in the work. Other specific comments are in the following:

  • A large portion of the introduction and materials and methods sections are directly copied from the authors earlier works published in Drug design, development and therapy (ref 32) and International journal of pharmaceutics (ref 50). They should be re-written to avoid self-plagiarism.
  • The key point on “effect of surface morphology” was not clearly demonstrated in the study. SD60, SD80 and SD100 were produced. While SD60 and SD80 have similar particle size, SD100 is significantly larger. In addition, the difference in surface morphology between SD60 and SD80 is not obvious. As a result, it would be difficult to identify if the difference in behavior is due to particle size or surface morphology.
  • In the description of PIV experiment, it is not clear what X-axis and Y-axis are referring to. It would be good to have the coordinate indicated in Fig 1(A).
  • The report of both XRD and DSC results are redundant. Both results indicate an amorphous structure after spray drying. Unless additional information is given, only one, either XRD or DSC results should be reported.
  • The PIV results can be considered quite valuable. It is disappointing that only information drawn from the PIV results is that SD60 and SD80 provide wider flow fields, velocities and directions than SD100. This discussion should be expanded. The current PIV images did not serve the purpose of proper comparison among the formulations. The authors can consider providing zoomed in images at various sections for better comparisons.
  • Essentially, the conclusion drawn that “particles with small particle size and high roughness would provide an improved deposition in the deep lung region for pulmonary delivery due to the smaller aerodynamic diameter and aggregate strength as a result of their surface morphology confirmed by PIV and CI” is not clearly justified based on the results reported. The effect of particle size and surface morphology has also been reported in many previous studies. The authors should give a better differentiation of the current work compared to those in the literature.

Reviewer 3 Report

The major problem of this paper is the organization of the text. The authors have the tendency to repeat many times concepts that are part of the knowledge of the researchers in inhalation.

In particular the part dedicated to dissolution of powder suffers from several mistakes and the analysis is not correct.

The most interesting part is the one dedicated to the results of velocimetry determination that are not efficiently presented. I suggest to the authors in the revision to anticipate the aerodynamic assessment and discuss the physical parameter in support to the respirability. Do not invent particle fine dose or fraction. Stay on the compendia. The bargraph is more informative if the Y axis present the mass instead of fraction.

I attached the revised pdf for helping the authors in the revisions.

English is very poor.

Round 2

Reviewer 2 Report

The revision is satisfactory and the responses are adequate.

Reviewer 3 Report

The authors introduced the most parts of the modification suggested. Now the manuscript can be accepted.

This manuscript is a resubmission of an earlier submission. The following is a list of the peer review reports and author responses from that submission.

Round 1

Reviewer 1 Report

The paper "The effect of surface morphology of spray-dried bosentan microparticles on aerodynamic performance for dry powder inhalation" focuses on a very interesting and hot aspect in the field of pulmonary drug delivery, namely the study of the relationship between process parameters in production of engineered microparticles, their physical characteristics and aerodynamic behavior. The authors approach the topic through the use of standard techniques and also with the use of an advanced particle image velocimetry technique.

In general, the work appears rather superficial, with some macroscopic errors and naivety in the collection of data that make them substantially unreliable. As a consequance, the discussion and conclusions appear incorrect.

Specific comments.

Lines 78 and 79: the meaning of the sentence is not clear.

Line 95: the word "parameters" is missing.

Line 109: please  indicate the typical size of the batches of powder produced.

Line 110: the table does not show "different amounts of bosentan".

Line 112 and table 1:  given that the solvents used had a different boiling point, it is reasonable to think that the outlet temperatures were different, while the authors indicate the same range (55-65 ° C) in all three cases studied. Please indicate the actual outlet temperature for each case.

Line 124: the DSC traces were recorded using sealed crucibles without pierce. This prevents the residual humidity present in the powder which evaporates due to the effect of heating from being released thus determining an increase in pressure inside the crucible which causes an "explosion" observable by the endothermic spikes in the traces of figure 3 between 40 and 70 ° C. All this makes the measurement inaccurate and reliable.

Lines 131-133: please provide more details about FTIR experiments.

Line 136: please provide details about the dimensions of the Franz cell (diameter of the permeation area, volume of the receptor compartment, whether the donor contained a small amount of liquid).

Line 145: why samples were centrifuges rather than filtered?

Line 146: dissolution tests were conducted on 3 replicates. In this way the calculation of the value of f2 does not appear correct. f2 is in fact based on a statistical approach that requires good data strength and robustness. The guideline cited indicates, in fact, that the analysis must be carried out on at least 12 replicates.

Lines 153-166: this approach is useless and substantially wrong. The equations used are designed and proposed to describe release kinetics (typically from matrix type systems) rather than a dissolution process. A Hixon-Crowell cube root model would have been more correct, or perhaps even better, the comparison between the dissolution rates could be done with a model independent approach such as the one using the Dissolution Efficiency or the Mean Dissolution Time.

Line 196: did the authors checked the pressure drop of 4kPa over the inhaler before the test?

Table 2: the description of the size distribution measure is completely missing in the experimental part. It is understood that the authors used a Dynamic Light Scattering equipment, which, however, does not seem to be the most suitable instrument for measuring the particle size distribution of these powders. The Poly Dispersity Index values (approximately equal to 1 in all cases) indicates that these measurements are totally unreliable and cannot be used. The discussion of the zeta potential is useless for particles of this size.

Line 287: I cannot see any “broad endothermic peak” but rather endo-eso phenomena for all the 3 SD powders, likely suggesting a glass transition of the amorphous phase.

Lines 283-285: the comment on the endothermic peak recorded for bosentan raw material requires a more in-depth assessment: in fact, contrary to what the authors said, the data relating to the peak at 113 ° C is not in accordance with what is reported in ref. 49 (Dangre et al. report 126 ° C). In any case, the reasonable possibility that this is the dehydration peak of the hydrated form has not been considered. In literature there are data on the melting point of bosertan at 107-110- ° C and at 195-198 ° C. This point therefore deserves further study also in relation to the physical stability of the SD powders, a problem, the latter, which was not minimally considered by the authors.

Line 306; no instead of to.

Figure 5 and relevant discussion have to be reconsidered. Please focus on the spectra region 1800-1000 cm-1 with better magnification and comment in more detail the differences between the raw material and SD samples.

Dissolution Studies lines 310-337:

The dissolution data collected refer to dissolved percentages lower than 30%, too low to discuss differences between the curves.

Regarding Table 3 and the related discussion, please delete them for the above-mentioned reasons.

Figure 7 and 8: please explain the meaning of the color scale.

As for the particle Imaging Velocimetry I totally disagree with the Authors interpretation and data discussion.

In my opinion, the differences in aerodynamic behavior are not so much attributable to the differences in morphology and surface characteristics, but rather to the particle size distribution. The images shown in the three videos in the Supplementary are quite clear in this sense as well as the data of Figures 8 and 9. The SD80 powder appears to be composed of both very large, fast moving particles and very small particles that tend to float in the air stream. The SD 100 powder appears to consist of larger and more uniform particles, moving quite rapidly, while the SD60 powder appears to have an intermediate behavior. From this point of view the authors should see the work of De Ascentiis et al. (Pharm. Res., 5, 734-738, 1996). Unfortunately, the authors have not measured the particle size distribution of the powders correctly and therefore they cannot make an adequate correlation.

The data shown in figure 10 seem to disagree with those of table 4 as regards the MMAD values of raw material and SD 100. The figures show that the ram material is distributed almost entirely in stages -1 and zero and is not clear, therefore how it can have a MMAD lower than SD 100. A similar argument also applies to FPF data. In any case, the GSD values of SD100 and raw bosentan indicate that the data have not been correctly evaluated. Probably the authors did not previously check that the distribution was log-normal.

Reviewer 2 Report

The manuscript has not sufficient novelty and originality to be published in Pharmaceutics. The AA have already published a very similar research. Therefore I recommend rejection.

Reviewer 3 Report

The work from Kwon Yong-Bin and co-workers concerns the preparation and the characterization of Bosentan hydrate (BST) microparticles by spray-drying, in order to obtain a formulation that could potentially be used for the treatment of moderate pulmonary arterial hypertension. However, the present work only deals with the selection of the best formulation, with the most suitable properties.

Generally speaking, the work has a good design on its basis despite the results and their discussion should be improved in order to make the paper more fluently in the reading. Indeed, the strictly separation and, thus, the discussion of each performed experiments (divided by type of techniques) reduce the connection and the natural correlation between the results. For this reason, I feel to suggest to the Authors to review the structure that has been given to the paper, maybe giving a uniform discussion.

Furthermore, Authors would like to consider the following suggestions:

  • Line 50: the acronym DPIs is explained at line 57; I suggest to explain the meaning of DPIs the first time it is cited (namely line 50).
  • Line 77: it is supposed that “variability” should be “variable”.
  • The sentence at lines 78-79 is not clear. Can the Authors consider to re-phrase it?
  • The same for sentence at lines 85-87.
  • Section 2, Materials and Methods: Authors should report how the size distribution of microparticles (referring to the results reported in Table 2) was performed, including the measurements of the zeta potential.
  • Line 112: which type of nozzle has been used?
  • Table 1: why only modification in BST solution was considered in order to produce particles with different properties?
  • Section 2.4: which reference material was used for the DSC analysis? Was the scanning performed under an inert atmosphere?
  • Section 2.6: How many scans were performed?
  • Section 2.10: How about the calibration curve? How many points were considered and in which range of BST concentration?
  • About the particle morphology (section 3.1), how do the Authors state that the diffusion rate of formulations SD100 and SD80 is higher than SD60? It is not clear if the considerations about the Peclet number, for which no value is reported, still remain theoretical or not. Furthermore, the information about the BST solubility in water and ethanol should be placed earlier in this section, in order to make the discourse more logical. I feel to suggest to the Authors to re-arrange this part.
  • Lines 255-256: since in these lines Authors gave comments on the size distributions of microparticle formulations, I suggest to the Authors to consider to place at this point of the manuscript the Table 2. For the reader, it is not so easy to follow the discussion produce about the particle size distributions without any presentation of this results.
  • Lines 268-271: why do the Authors comment at this stage of the manuscript the results from Table 4, that is presented much later in the discussion? Honestly, this creates a little bit of confusion.
  • Table 2 should report a size distribution as explained in the caption of this table. However, Authors just reported the Z-average diameter. I fell to suggest to the Authors to report more details about the distribution of the size of all formulations.
  • How many analyses were performed for the size analyses?
  • Line 278-280: are there any information from literature that confirm the negative zeta potential value obtained?
  • Line 289: what is a “solvent with high solubility”?
  • Lines 306-307: what is reported in these lines is maybe a typo error, since probably Authors want to state that no chemical changes occurred.
  • Figure 5: Can the Authors give a comment on the peak at about 3500 cm^-1 present in raw BST that is not more visible in the case of all spray-dried formulations?
  • What about the few variations in the intensity in some FT-IT peaks in the range between 500 and 1500 cm^-1? Can they have a statistical significance difference with the spectrum of raw BST?
  • Line 320: Authors stated that the amorphous solid state increases the solubility of BST. However, since the final amount of BST dissolved is the same from raw BST and BST spray-dried, was the Authors referring to the increment in the dissolution rate rather than the solubility itself?
  • Line 331: Authors stated that “[…] the release rate showed similar dissolution constants” referring to the Higuchi model. Does this mean that there were no statistical differences in the dissolution constants?
  • In Table 3, looking at R^2 for SD100, SD80 and SD60, higher values were obtained for Korsmeyer-Peppas fitting. However, Authors commented that, based on the n-value, the release is anomalous and follow Higuchi model. Can the Authors give a deeper comment on this?
  • Lines 380-381: this sentence is not clear. Due to the spherical shape and smooth morphology of the particles, does the dynamic shape factor equal to 1?
  • Line 382: what is higher? The subject of this sentence is not clear, can the Authors rephrase?
  • Table 4: what is RF? It is never defined in the manuscript.
  • Section 3.5: it is opinion on this reviewer that the presentation of the results in this section is not particularly clear and the discussion is quite weak. Considering that the aim of the BST spray-dried formulation is a delivery of the drug to the lung, this section is expected to be one of the most important. For this reason, I fell to suggest to the Authors to implement this part.
  • Line 418-419: the sentence in these lines is a repetition of what was stated in previous sentence.
  • Throughout the manuscript there are some typo errors (e.g. lines 254, 272, 334, in Table 3 “N” should be in lowercase, 342, 364, 369, 373,